# Effects of Air Pollution on Cellular Senescence and Skin Aging

**DOI:** 10.3390/cells11142220

**Published:** 2022-07-17

**Authors:** Ines Martic, Pidder Jansen-Dürr, Maria Cavinato

**Affiliations:** 1Institute for Biomedical Aging Research, Universität Innsbruck, 6020 Innsbruck, Austria; ines.martic@student.uibk.ac.at (I.M.); pidder.jansen-duerr@uibk.ac.at (P.J.-D.); 2Center for Molecular Biosciences Innsbruck (CMBI), Innrain 58, 6020 Innsbruck, Austria

**Keywords:** senescence, air pollution, skin aging, cosmetics

## Abstract

The human skin is exposed daily to different environmental factors such as air pollutants and ultraviolet (UV) light. Air pollution is considered a harmful environmental risk to human skin and is known to promote aging and inflammation of this tissue, leading to the onset of skin disorders and to the appearance of wrinkles and pigmentation issues. Besides this, components of air pollution can interact synergistically with ultraviolet light and increase the impact of damage to the skin. However, little is known about the modulation of air pollution on cellular senescence in skin cells and how this can contribute to skin aging. In this review, we are summarizing the current state of knowledge about air pollution components, their involvement in the processes of cellular senescence and skin aging, as well as the current therapeutic and cosmetic interventions proposed to prevent or mitigate the effects of air pollution in the skin.

## 1. Introduction

The skin is the largest and outermost organ of the human body. As such, the skin represents the major protective barrier between the internal and external environment and protects the body from environmental damages. Additionally, the skin is important for the regulation of body temperature and water loss and participates in certain immune responses [1].

Skin aging is a result of two cumulative and overlaying mechanisms denominated as intrinsic and extrinsic aging. The process of intrinsic or chronological aging affects all tissues and organs of the body, is due to the passage of time, and is influenced by genetic background. The main signs of intrinsic aging in the skin are relatively mild, including the accumulation of fine wrinkles with moderate changes in skin pigmentation [1]. In addition, the skin is continuously exposed to environmental and lifestyle factors such as sunlight, pollution, cigarette smoke, and dietary habits. These factors, collectively denominated the skin exposome, are the major causes of the process of extrinsic skin aging. Major characteristics of extrinsic skin aging are coarse wrinkles, solar elastosis, and pigmentation irregularities [1,2]. Quality of life as well as emotional well-being are influenced by changes in the skin appearance [1,3].

From all factors of the exposome, sunlight is known to be the most harmful and for this purpose extrinsic skin aging is also referred as photoaging. Arguably, chronic exposition of the skin to sunlight leads to skin aging, which is characterized by increased oxidative stress, apoptosis, stimulation of melanogenesis and direct damage to DNA, membranes, and proteins [1,3,4].

A potentially serious, yet less recognized environmental factor is ambient (outdoor) air pollution. As a result of rapid industrialization and urbanization, environmental pollution is becoming a severe public health issue worldwide [5]. In 2019 the World Health Organization (WHO) determined that 99% of the world’s population lives in places where air pollution levels exceed WHO limits and thus air pollution was designated as the world’s largest single environmental health risk to humans [6]. Pollution has been strongly correlated to the degenerative processes of skin aging, particularly pigmentation issues, as well as to the onset of skin disorders [7,8].

Different compounds can be categorized as air pollutants, and as such each contribute differently to the impairment of the skin [9]. These compounds differ in chemical composition, reaction properties, ability to diffuse, and time of disintegration [10]. Given that the interest in the effects caused by exposure to air pollution has risen in the last few years, the number of publications about this topic is constantly increasing. Recent studies show that air pollutants such as particulate matter (PM) can cause skin tanning, skin aging [11,12], and inflammatory skin diseases such as atopic dermatitis (AD), acne, and allergic reactions [13]. Further investigations are needed to fully understand how skin is affected by the exposure to environmental stressors. Given that the skin is not exposed to one single source of environmental impact, it is especially important to examine how combined effects (e.g., UV light, air pollution, cigarette smoke) of different stressors damage the skin and contribute to aging of this tissue. 

In this review we summarize the different components of air pollution and how they contribute to skin aging and cellular senescence. Furthermore, we will demonstrate how accumulation and persistence of senescent cells impact the process of skin aging. Additionally, we outline current interventions and cosmetic compounds used to prevent and mitigate effects of air pollution in the skin (Figure 1).

## 2. Cellular Senescence

Cellular senescence, which is considered one hallmark of aging, is defined by a state of proliferative arrest and is known to be involved in tumor suppression and progression, tissue remodeling, and embryonic development [14]. The accumulation and persistence of senescent cells is an important characteristic of aging of some tissues, including the skin [15]. Senescent cells contribute to the decline of tissue function and lead to age-related changes and pathologies [16]. This well-known interconnection between cellular senescence, skin aging, and skin diseases is often related to molecular processes such as inflammation (Figure 1).

Senescent cells are characterized by several characteristics such as a distinct morphology [17], DNA damage [18], cell cycle arrest [19], mitochondrial dysfunction [20], protein quality impairment [21], inflammation [22], and reactive oxygen species (ROS) generation [23]. Additionally, senescent cells are not able to proliferate as they reside in a cell cycle arrest, which is mostly caused by an accumulation of DNA damage [24,25]. If unrepaired, these damages lead to detrimental effects such as cellular dysfunction and cancer. Other features of senescent cells are the appearance of senescence-associated heterochromatin foci (SAHF) and decreased Lamin B1 expression [14].

Importantly, some types of senescent cells can also be recognized by other characteristics, such as increased senescence-associated β-galactosidase (SA-β-Gal) activity and the secretion of a set of pro-inflammatory factors, the so-called senescence-associated secretory phenotype (SASP) [26]. In general, SASP components can have beneficial effects such as recruitment of immune cells, promotion of anti-tumor response, and improved wound healing [26]. On the other hand, SASP components can contribute to the functional disruption of tissue structure in an autocrine and paracrine manner leading to the senescence of the neighboring cells via paracrine communication, immunosuppression, and inflammation [26,27]. The composition of the SASP is determined by the type of senescence and cell type but the most common cytokines upregulated in cutaneous skin cells during aging are interleukin (IL)-1 and IL-6, as well as the matrix metalloproteases (MMP)-1 and MMP-3 [26,28].

Another hallmark of senescent cells is the imbalance between their production of ROS and their ability to detoxify the reactive intermediates [29,30]. Elevated ROS levels cause damage to cellular molecules such as proteins, lipids, and nucleic acids as well as to organelles such as mitochondria and proteasomes [31]. Antioxidant enzymes such as glutathione and superoxide dismutase are able to stabilize or deactivate free radicals before they attack cellular components, avoiding the so-called oxidative stress. During senescence these defense mechanisms are decreased whereas the ROS is continuously produced and the cells become more prone to oxidative damage [32]. ROS can induce the mitogen-activated protein kinase (MAPK)-p38 pathway which leads to further activation of p53-p21 and cell cycle arrest and thus initiating or accelerating the senescence process [14]. Besides this, excessive ROS induce inflammation-related pathways such as the nuclear factor kappa-light-chain-enhancer of activated B cells (NF-κB) pathway, which regulates different intracellular responses such as apoptosis, cell proliferation, angiogenesis, metastasis, and tumor promotion [7].

Mitochondrial damage, another feature of senescent cells, results from accumulation of mitochondrial DNA mutations, dysfunction and structural alterations of important mitochondrial proteins and membranes, imbalance of fission and fusion, and impaired mitophagy which, in turn, results in increased ROS production and decreased energy generation by oxidative phosphorylation [20,33]. Mitochondria are the main intracellular sources of ROS and excessive ROS production can lead to fragmentation of mitochondria by modulation of mitochondrial fission and fusion proteins [34] and impairment of mitostasis is considered to be one of the causes of cellular senescence [20].

During senescence, the intracellular mechanisms of protein quality control, autophagy and the proteasome, are impaired. Autophagy is responsible for the elimination of damaged or excessive proteins and organelles via the lysosomes, whereas the proteasome especially degrades oxidized proteins [35]. Generally, the activity and function of both mechanisms are reduced with age and senescence [21,24]. For instance, decreased expression of genes that transcribe proteasome subunits as well as modifications of these subunits are considered as major drivers of the age-related impairment of proteasome activity [36]. Proteasome activity has been reported to be decreased in photoaging models of keratinocytes [37], melanocytes [17], and fibroblasts [31]. Furthermore, in senescent melanocytes and fibroblasts the impairment of the proteasome was compensated by an increase of autophagy as an alternative mechanism of protein quality control [17,31]. Impaired autophagy has been related to reduction in lysosomal proteolytic function, decreased rates of autophagolysosomal fusion, and impaired delivery of autophagy substrates to lysosomes, which leads to the intracellular accumulation of undigested material, exacerbates cellular impairment, and contributes to the development of senescence and in tissue to age-related diseases [21,38,39].

## 3. Cellular Senescence and Skin Aging

### 3.1. Main Characteristics of Skin Aging

The skin is constituted of many different cell types, including, among others, fibroblasts, keratinocytes, and melanocytes. Skin aging is a multifactorial process and most if not all skin cell types, when functionally impaired, can potentially contribute to the deterioration of the tissue [15]. Additionally, the skin is a useful system for the investigation of aging since this tissue undergoes morphological, biochemical, and functional modifications during the process of aging. 

Skin aging is characterized by different features, one of which is the accumulation of damaged and dysfunctional macromolecules in the skin cells due to decreased function of autophagy and proteasome activity. Autophagy plays a role in extrinsic and intrinsic skin aging and regulates pigmentation, homeostasis, and the functions of fibroblasts, keratinocytes, and melanocytes [21,39]. In fibroblasts, it was shown that impaired autophagic flux induced by inhibition of lysosomal proteases lead to the decreased expression of hyaluron, elastin, and type 1 procollagen, and to the increased breakdown of collagen fibers which result in the impairment of dermal integrity and increased skin fragility [40]. In melanocytes, several groups showed that modulation of autophagy can induce cellular senescence and impairment of antioxidant defense mechanisms leading to changes in skin pigmentation [41,42]. Additionally, autophagy and oxidative stress are determinant factors in the fate of keratinocytes and control their progression to senescence, programmed cell death, or tumor formation [21,39].

During aging, skin cells also accumulate damaged mitochondria and mitochondrial DNA deletions leading to structural and functional alterations in the extracellular matrix (ECM) and the induction of inflammation which, in turn, accelerate the formation of skin wrinkles [18]. 

### 3.2. Inflammation and Skin Aging

Inflammation is one of the main pathways involved in the process of skin aging. Furthermore, chronic inflammation can lead to skin disorders such as AD and psoriasis [43]. AD is a skin inflammatory disease with symptoms such as pruritus, itching, pain, and sleep disturbance [44,45]. Several different pathways are involved in the activation of inflammation. Here, we will focus on three main activators of inflammation recognized to be involved in skin aging.

The first pathway involves the aryl hydrocarbon receptor (AhR), a transcription factor highly expressed in all cutaneous cell types which has been reported to play a key role in the maintenance of skin barrier, regulation of skin pigmentation, and skin immunity. In humans, the highly conserved AhR has a complex role. Different ligands such as polyaromatic hydrocarbons (PAH) can bind and activate AhR. This leads to AhR translocation to the nucleus where it regulates the expression of genes such as cytochrome P450 1A (*CYP1A*) which can, in turn, further induce oxidative damage by generating ROS [46]. Alterations in AhR signaling lead to dysregulated skin barrier function and can generate symptoms such as dryness, itchiness, and flakiness which are similar symptoms to AD [46,47,48]. In mice, depletion of AhR leads to transepidermal water loss, decreased expression of important barrier function proteins such as filaggrin and involucrin, and changes of the skin microbiome [49]. In chronic skin inflammation diseases such as psoriasis, activation of AhR can determine the severity of the symptoms [50]. In addition, the activation of AhR is crucial for melanocyte survival and melanogenesis which are events that can be linked to the appearance of senile lentigines [51]. 

The second important inflammatory pathway participating in skin aging is controlled by NF-κB, a transcription factor which resides in the cytoplasm and, if activated, translocates to the nucleus. Activation of the NF-κB signaling pathway is driven by the response to diverse stimuli, including ligands of various cytokine receptors, pattern-recognition receptors (PRRs), T-cell receptor (TCR) and B-cell receptors, as well as to a subset of TNF receptor (TNFR) superfamily members such as LTβR, BAFFR, CD40, and RANK [52]. Once translocated to the nucleus, NF-κB promotes the production and release of tumor necrosis factor-α (TNF-α), MMPs, and other SASP components such as IL-1α, and cyclooxygenases (COX)-1 and -2. These factors can reinforce the inflammation process and accelerate the skin aging process [53,54]. Importantly, the dysregulation of NF-κB plays an important role in the pathogenesis of chronic inflammatory diseases of the skin as well as in wound healing [55]. 

The third important inflammatory signaling pathway involved in skin aging is coordinated by nuclear factor erythroid 2-related factor 2 (Nrf2)/antioxidant response element (*ARE*). Nrf2 is an important transcription factor in the inflammation signaling cascade as well as in oxidative stress responses and is responsible for the expression of *ARE* genes such as heme oxygenase-1. Nrf2 can also negatively regulate NF-κB activation either by directly decreasing intracellular ROS levels or by inhibiting the translocation of NF-κB to the nucleus [56,57]. Recently, it was demonstrated that senescent melanocytes increased the expression of Nrf2 which was accompanied by decreased melanogenesis [58]. Nrf2 depletion in skin cells leads to the reduction of cell survival as well as induction of oxidative stress while mutations in the *Nrf2* gene, in turn, lead to the development of squamous cell carcinoma suggesting that pathways activated by Nrf2 are important for the maintenance of skin homeostasis [59].

### 3.3. Senescence of Skin Cells and Skin Aging

The accumulation of senescent cells in the epidermis and dermis is considered as a hallmark of aging. The occurrence of senescent cells can be accelerated by exposition of the skin to different sources of environmental and lifestyle factors [15,60]. Senescent fibroblasts display different senescence associated characteristics such as cell cycle arrest, decreased autophagy activity, increased SA-β-Gal activity, mitochondrial dysfunction, DNA damage, increased ROS generation, and increased expression of SASP factors [31,61]. The appearance and persistence of senescent fibroblasts in the dermis leads to the accumulation of elastotic material, which results from the incomplete degradation of elastic fibers, as well as to the decreased expression of extracellular matrix components simultaneously to the increased degradation of ECM. These events are manifested in dullness and reduced elasticity of the skin [1,62]. Keratinocytes are constantly renewed and more prone to apoptosis than senescence when confronted by stressors. Nevertheless, keratinocytes contribute greatly to the aging process by losing the ability to terminally differentiate or proliferate and responding differently to external stimuli [15]. Senescent keratinocytes express increased p15, IL-1α, and high mobility group A2 [27]. Other hallmarks of senescence such as SA-β-Gal, expression of p21, p53, and p16 are displayed in keratinocytes upon UVB exposure [16]. Senescence of melanocytes has been insufficiently explored but a recent study has reported that senescent melanocytes accumulate in human skin where they contribute to aging of this tissue by impairing proliferation of the neighboring keratinocytes [63]. Melanocytes’ senescence can occur either by exhaustion of their replicative capacity, in a process called replicative senescence, or by stress-induced premature senescence in which senescence is triggered by chronic exposition of these cells to sublethal doses of a stressor agent such as UV or ROS-inducing agents [17,25]. Accumulated senescent melanocytes in skin is correlating with visible signs of skin aging such as facial wrinkling and deposition of elastin in the dermis [60]. Although melanocytes become age-dependently less active, darker pigmented spots, also described as age spots, solar lentigines, or lentigo senilis, are a common characteristic of aged skin, presumably occurring due to irregularities in melanocyte distribution, enhanced melanogenic signaling and decreased melanosome removal [15,64]. Besides the impairment of melanocytes, the dysregulated secretion of molecules produced by aged keratinocytes and fibroblasts cause the occurrence of senile lentigines [15,51]. For example, UV-induced senescent fibroblasts contribute to the formation of age spots by decreasing the expression and secretion of stromal cell-derived factor 1 [65], emphasizing the complexity of underlying factors driving skin aging and the development of senile lentigines. 

### 3.4. Microbioma and Skin Aging

Another important but as yet less investigated change that occurs in the skin during aging is the alteration of the skin microbiome. The human skin microbiome consists of bacteria, fungi, viruses, archaea, and other microorganisms which are essential for body homeostasis and when dysregulated can contribute to diseases [45,66]. The skin of young individuals is rich in bacteria of the Firmicutes phylum but with age, these are replaced by Bacteroides and Proteobacteria. This imbalance of commensal skin microbes which produce immune factors affects the skin’s immune system, explaining the increased risk of pathogenic invasions and age-related skin disorders [22,67]. 

### 3.5. Skin Aging in Different Ethnicities and Phototypes

In humans, melanin determines skin and hair color and is responsible for photoprotection against UV radiation. Melanin molecules surround the nuclei of the keratinocytes and melanocytes to protect their genetic material, by absorbing sunlight through the polymeric melanin molecule [68,69]. The differences in skin color are mainly determined by the type and amount of melanin produced by melanocytes which is a major factor of skin aging. Melanocytes can produce two different types of melanin. Eumelanin, a brown-black pigment, and pheomelanin, a yellow-red pigment [70]. Melanocytes transfer melanin to the neighboring keratinocytes, and melanin is degraded during keratinocytes differentiation. The ratio of melanin degradation determines the skin pigmentation and, consequently, the skin phototypes. For instance, in fair skin phototypes melanosomes are almost completely degraded, whereas in dark skins they barely undergo degradation and accumulate in the upper layers of the epidermis [1]. During the process of skin aging, the density and activity of melanocytes is continuously decreased, while the percentage of cells with impaired melanin synthesis is increased [15], leading to the appearance of skin pigmentation issues.

Given the differences in skin pigmentation and degradation of melanin, skin aging manifests differently among different skin phototypes and ethnicities. For instance in Asians and African Americans pigmentation changes are considered early manifestations of skin aging, whereas aging of Caucasian skin is characterized in its early stages by the appearance of wrinkles [71,72,73].

In general, highly pigmented skin structurally consists of a thicker dermis and stratum corneum which contribute to increased resistance and elasticity in comparison to light pigmented skin [71,74]. Recent studies demonstrated that highly pigmented skin exposed to UV displays a deposition of collagen and elastic network, a decrease in epidermal thickness [75], lightening of skin [76], as well as production of MMPs [77]. Upon skin aging, light pigmented skin exhibits thinner and less compact stratum corneum, deposition of elastic fibers, impairment of barrier function, reduction of sebum, production of MMPs, and loss of collagen [71,74]. These findings indicate that skin aging mechanisms seem to be similar between ethnicities. However, further investigations are necessary to understand the underlying molecular processes of different ethnic skin types.

## 4. Major Components of Air Pollution Affecting Skin Appearance

Pollution is defined as an environmental contamination by chemical, biological, or physical substances which can affect human health and ecosystems. Air pollutants are composed of organic and inorganic substances which are introduced into the atmosphere by residential wood heating, tobacco smoking, transportation, and industry, among other sources [78]. The composition of atmospheric pollution can also vary depending on the time of day, seasons, human activity, and geographic location [8,9]. Several reports suggest that air pollutant components are direct contributors to the process of aging [5,11,79,80,81,82]. Human exposition to air pollution contributes to increased mortality and hospital stays. The effects of air pollution can range from nausea, difficulty in breathing, skin irritation, birth defects, and reduced activity of immune system, to cancer. Results obtained from research with animal models and epidemiological studies suggest that the cardiovascular and respiratory system are the main affected systems by air pollution [10]. In the lungs, infiltration of air pollutant components, especially small particles which can reach bronchial tubes and deep lung, can induce a systemic immune response due to increased expression of IL-1, IL-6, IL-8, and monocyte chemoattractant protein-1 in macrophages and lung epithelial cells, leading to the development of respiratory diseases [5].

The skin can be affected by air pollution in two ways. Either directly, through the uptake of the air pollutants by the skin, especially by intrusions formed by hair follicles in the stratum corneum [5], or indirectly by the uptake of particles by the lungs which are further transported by the blood to the skin [9]. 

Exposure of the skin to urban air pollution activates mechanisms of cell detoxification, which, if active over a longer period of time, can lead to DNA and protein damage, elevated ROS levels and lipid peroxidation, resulting in skin alterations, such as impaired barrier function, pigment spots, wrinkles, and decreased skin hydration [83,84]. Additionally, air pollution induces inflammation, activates the AhR pathway, and leads to alterations of the skin microbiome [85,86]. Given the strong link between low-grade systemic inflammation and biological aging, it is possible that exposition of the skin to air pollution leads to premature cellular senescence and, consequently, to skin aging [85]. The literature regarding the effects of air pollution on different ethnic skin types is still elusive. The few broad ethnic studies show that exposition of the skin to air pollution mainly induces increased oxidative stress leading to skin pigmentation disorders and wrinkle formation [2,11,72]. The majority of the publications are exclusively related to Asian skin [87] and therefore very biased. Information on the effects of air pollution on different skin types is still lacking and therefore the awareness of different ethnicities in this field of research needs more attention.

In this review we will focus on the most investigated and relevant air pollutants for the process of skin aging: ozone, heavy metals, cigarette smoke, and PM.

### 4.1. Particulate Matter

Particulate matter (PM) is one of the main components of air pollution and is defined as a mixture of gas containing liquid and/or solid droplets varying in size and composition [86,88]. Particles contained in PM include substances such as metals, minerals, organic toxins, tobacco smoke, pollen, allergens, and smog [89]. These particles are classified according to their aerodynamic diameter (Dp) in three categories: particles with Dp bellow 10 µm are named “coarse particles” or PM 10 and include components of dust, soil, and dusty emission from industries; particles with Dp between 0.1 and 2.5 µm are called “particle matter” or PM 2.5 and are mainly derived from open fires, automobile exhausts, and power plants [90]; and particles with Dp bellow 0.1 µm are called “ultrafine particles” or PM 0.1 and mainly consist of emissions of diesel-powered engines [86]. The aerodynamic diameter of the particles is an important determinator of their ability to enter the body by alveolar–capillary barrier and travel across the blood [91]. The particle matter and ultrafine particles can penetrate the body either by systemic distribution through the blood circulation after entering the lungs’ alveoli or by infiltrating the skin through hair follicles [92]. Recently, in contrast with what was believed before, it was demonstrated that coarse particles can also penetrate the stratum corneum or the respiratory system [53,93]. 

Exposure to particulate matter can cause cardiovascular and respiratory diseases, allergies, and cancer through different mechanism [83,94]. For instance, PM 2.5 exposure induces endoplasmatic reticulum stress response and apoptosis in the lung and liver of mice and this mechanism is believed to be one possible explanation for the development of metabolic, respiratory, and cardiovascular diseases in humans exposed to air pollution [95]. A study in mice showed that PM inhalation activates the TNF-α driven systemic inflammation and results in an impaired cardiac function which to a certain extent was prevented by a TNF-α inhibitor called inflimiximab [96]. In skin diseases such as psoriasis, the blockage of TNF-α leads to detrimental side-effects such as increased risk of infections and malignancies as well as to the formation of new and more psoriatic skin lesions [97]. 

A cross-sectional study which investigated the correlation of particulate matter exposure and aging demonstrated that leukocytes of peripheral blood from elderly humans exposed to daily high concentrations of PM 2.5 presented decreased telomere length, lower mitochondrial DNA content, and reduced sirtuin-1 expression [98]. In another study it was shown that exposure of human nasal epithelial cells to PM 2.5 leads to ROS production, degradation of tight junction proteins such as occludin, claudin-1, and E-cadherin, leading to sinonasal diseases through disrupted tissue integrity and permeability [45,99]. Similar downregulation of tight junctions as well as keratins and filaggrin were observed in pork skin in response to PM 2.5 exposure, leading to increased skin permeability [100,101]. Altogether, these studies demonstrate that short- and long-term exposure to PM can induce features of premature aging.

Particularly in the skin, PM 2.5 induces different detrimental processes such as DNA damage and lipid peroxidation. Exposure of the skin to PM results in the formation of senile lentigines, increased formation of ROS, and promotes the release of pro- inflammatory cytokines which all lead to accelerated skin aging and increased susceptibility to pathogen invasion [5]. ROS generation, induced by exposure of the skin to air pollution, can induce MMPs expression and increase their activity, especially MMP-1 and MMP-3 which are known to accelerate the skin aging process by degrading collagen and elastin. Additionally, the decreased expression of transforming growth factor (TGF) β, and reduced synthesis of collagen type 1 α chain (*COL1A1, COL1A2*) and elastin by fibroblasts are other contributors to the formation of wrinkles and skin aging induced by PM [80,82,102]. 

In human keratinocytes and mouse skin tissue, PM induced the expression of demethylases such as DNA demethylase 1 (TET1) and decreased the expression of DNA methylation-related proteins such as (DNMT)-1 and -3. These changes in the methylation pattern of DNA induced “skin senescence phenomenon” are characterized by features such as hyperkeratotic epidermis accompanied by the increased expression of keratin-10, an epidermal differentiation marker, and proliferating cell nuclear antigen, a proliferation marker [79]. These results demonstrate that PM-induced skin aging can be triggered by epigenetic modifications which could give rise to new strategies for therapeutics against skin aging.

Skin cells exposed to diesel particulate extract (DPE), which mainly contains PM and PAH [103], displayed dysregulation of proteins and lipids important for the maintenance of skin integrity, for the regulation of skin hydration and oxidative stress, such as NADPH oxidase (NOX), ceramide, plakins, transglutaminases, cystatins, and filaggrin [57,80]. Furthermore, DPE impaired mitochondrial oxidative phosphorylation and cell migration in these cells. These effects could be partially avoided or restored by the treatment of the cells with the antioxidant vitamin E [57]. 

PM causes inflammation in the skin through increased IL-8, MMP-1, ROS production, and neutrophil infiltration in the deep dermis [104]. Fibroblasts cultivated with conditioned medium obtained from immortalized human keratinocytes (HaCaT) treated with PM displayed increased nuclear translocation of p65 and p50 as well as increased ROS production, morphological changes, and secretion of SASP components including prostaglandin E2, COX-2, TNF-α, IL-1β, and IL-6 [93]. It is well established that the expression and release of TNF-α is increased upon UV exposure [105] as well in some skin diseases such as vitiligo [106].

Other studies have shown that the exposition of HaCaT cells or reconstructed human epidermis to PM induced upregulation of NF-κB, COX-1 as well as IL-1α leading to skin barrier dysfunction [81]. In addition, an increased autophagic activity shown by the turnover of the light chain 3 Ι (LC3) to LC3 II, expression of p62, and PM internalization in the autolysosomes was observed after exposure of fibroblasts to air pollutants such as PM [107,108]. 

The PM particles can transport organic chemicals and metals into cells which when localizing in the mitochondria can directly generate ROS [2]. Mostly, PM particles contain PAH which can penetrate the skin. PAH is an activator of AhR in melanocytes and keratinocytes and exposition of the skin to this chemical can influence epidermal turnover, melanogenesis, and differentiation [9,86]. PAH have synergistic effect with UVA and both contribute to skin carcinogenesis and to the appearance of skin pigmentation disorders such as senile lentigines [5,53,70,92,109,110]. Interestingly, exposure of melanocytes to PM induces apoptosis through cytochrome C release and activation of caspase-3 [111]. These events are linked to the disappearance of melanocytes which is considered one of the main pathogenic mechanisms of vitiligo.

The effects of PM 2.5 and 10 are linked to different skin diseases such as AD and allergies [45,112]. Several studies have demonstrated that AD skin, as well as healthy skin exposed to PM, display skin barrier disruption due to decreased expression of epidermal structural proteins such as filaggrin, E-cadherin, and cytokeratins [43,44,113]. Additionally, it is suggested that AD can be influenced by the exposure to air pollution resulting in imbalance of immune cell response and IgE production, activation of AhR/NF-κB, and the generation of ROS, and these effects can be prevented by improved air quality [5,44,85].

### 4.2. Cigarette Smoke

Cigarette smoke is composed of different chemical substances including reactive oxygen species, carbon monoxide, and reactive nitrogen species. Smoking is associated with lung cancer as well as with cutaneous squamous cell carcinoma. PAH, one important component of cigarette smoke, is a major contributor to cancer. PAH induces AhR and subsequently *CYP1A1* which leads to the formation of DNA adducts and can result in cancer development [114,115].

The effects of cigarette smoke are cumulative and associated with the appearance of signs of premature aging, especially in the facial skin. Among them the most common are deep wrinkles, dryness, leathery texture, sagging, premature graying, and orange to purple discoloration of the skin [8,53,88]. Exposition of the skin to cigarette smoke induces oxidative stress and the impairment of the antioxidant system. The consequences are transepidermal water loss, lipid peroxidation, cell death, and degeneration of connective tissue by MMP-1 and -3 [80,114,116]. In keratinocytes, cellular redox homeostasis and colony-forming potential was affected in response to exposure to tobacco smoke components such as PAH and PM [92]. In humans, a study comparing smokers and nonsmokers revealed the occurrence of shorter telomeres, a sign of cellular senescence, in peripheral white blood cells of smokers [98]. Synergistic effects may occur upon the combined exposure to UV and cigarette smoke leading to epidermal barrier disruption, increased erythema, and decreased elasticity [8,53,88]. Furthermore, a three months’ whole body exposure of mice to tobacco smoke caused premature aging evidenced by the loss of collagen as well as hair loss and premature graying due to increased apoptosis and decreased melanogenesis, respectively [117]. Additionally, the combined exposure of tobacco smoke with ultraviolet light, exacerbated the effects of aging by the upregulation of p16 expression. These effects could be prevented by *N*-acetylcysteine (NAC) suggesting that premature aging triggered by cigarette smoke exposition is driven by ROS [5]. Other studies showed that exposure of skin cells, ex vivo skin biopsies, or reconstructed skin to cigarette smoke leads to production of pro-inflammatory cytokines and MMPs, lipid peroxidation, and downregulation of differentiation proteins such as loricrin and this, in turn, resulting in the impairment of skin barrier structure and function [114,115,118]. 

Besides the consequences of topical exposition of the skin to pollution, the inhalation of pollution has systemically effects to the skin. In a recent in vivo study, it was demonstrated that chronic inhalation of tobacco smoke leads to changes in the composition and deposition of elastin and fibrillin-rich microfibrils in the dermis which is accompanied by a higher stiffness of the skin [119].

In a study using a model of skin aging induced by tert-butyl hydroperoxide (tBHP), Wedel and colleagues showed that skin exposed to this chemical displayed downregulation of collagen synthesis and the increased secretion of MMPs. tBHP is categorized as an oxidative stress inducer which leads to the accumulation of intracellular ROS and deplete the antioxidant mechanisms and can be used as a proxy to study mechanisms that recapitulate the exposition of the skin to environmental stressors such as cigarette smoke [61].

### 4.3. Ozone

Stratospheric ozone has a protective role for earth-living organisms by filtering UV radiation. Ozone (O_3_) in the troposphere reaches the skin surface and reacts with molecules such as lipids and proteins in the stratum corneum [9,120]. In general, ozone causes oxidative stress, increased lipid peroxidation, AhR induction, and depletion of Vitamins C and E in human skin [121,122]. A report has shown that in human keratinocytes, exposure to ozone leads to lactate dehydrogenase release, reduced cell proliferation, lipid peroxidation, and NF-κB activation and these effects can be prevented by pretreatment with antioxidant mixtures [123]. Another study has shown that the combination of UVA and ozone has a synergistic effect and causes increased oxidative stress of the skin and Nrf2 expression in keratinocytes leading to skin inflammation, formation of wrinkles, and pigment spots [8,9,104]. In human skin O_3_ leads to activation of NF-κB, MMP-9, COX-2, and lipid peroxidation in the epidermis, whereas in the dermis expression of collagen-1 and -3 are downregulated. These consequences are accelerating the appearance of skin aging signs including the formation of wrinkles and senile lentigines as well as affecting wound healing [124,125].

### 4.4. Heavy Metals

The earth crest is formed by different components, among them heavy metals such as chromium, lead, cadmium, silver, nickel, mercury, manganese, and vanadium [8]. These metals can contaminate the water and food supply and cannot be degraded or destroyed. These substances are important trace elements for the maintenance of metabolic reactions but in higher concentrations they become toxic to the human body [10]. Heavy metals from industrial fumes reach plants through acidic rain. For instance, tobacco leaves are rich in cadmium which, when inhaled, cannot be excreted from the human body and leads to long term effects and damage to the lungs, kidneys, and bones [78]. A study showed that keratinocytes and skin explants exposed to dust particles containing heavy metals or heavy metals alone displayed increased expression of the pro-inflammatory cytokines IL-6, IL-8, caspase-14, and granulocyte macrophage colony-stimulating factor which are known to alter epidermal differentiation, ECM, apoptosis, DNA damage, lipid peroxidation, and skin immunity resulting in cutaneous inflammation and inflammation skin disorders [86,126].

Figure 2 shows a graphic representation of the complexity of air pollution damages to the skin and summarizes the most important signaling pathways regulated by different air pollutants on a cellular and on tissue level (Figure 2).

## 5. Therapeutics and Cosmetics

In recent years, due to increased industrialization and use of transport, the effects of air pollution on human health have increased logarithmically. As a consequence, a trend of development and use of anti-pollutant cosmetics and therapeutics originated in Asia, which is one of the most polluted places in the world. This trend spread to the western world and the demand in cosmetics and the personal care industry rose worldwide and led to increased research on the identification of new approaches and ingredients for anti-pollution cosmetics [127] (Table 1).

Several studies suggest different approaches to protect the skin from air pollution. Firstly, and most important, avoid exposure to air pollutants. Additionally, as the skin is exposed to air pollution and UV simultaneously, another measure to increase the protection of the skin is to use sunscreen, decreasing the possibility of occurrence of photoreaction [9,51]. Overwashing the skin can be harmful and break the skin barrier and homeostasis of the skin microbiome [9]. Indoor air ventilators or filters can also be helpful since they reduce the pollutant load in the air and on the skin [8]. One additional approach to protect the skin from external factors is the use of cosmetic preparations containing pro-, and prebiotics [128]. Components from our diet are also beneficial. UV-exposed skin revealed changes in the elastic and collagen network of the dermis including the composition and deposition of fibulin-5 which contributes to the structural formation of elastic fibers such as elastin. Additionally, the study showed that oral supplementation with green tea catechins in combination with vitamin C protected specifically fibulin-5 fibers against UV-induced changes [129]. Recently, reports have suggested that components from plants and other natural resources such as phenolic compounds, phytosterols, and saponins help against induced cellular oxidative stress by PM [10,13,85].

Recommendation of the topical use of cosmetics that protect or improve the skin barrier, enhance the antioxidant mechanism, and reduce inflammation can help avoid or mitigate air pollution skin damages [86]. Additionally, to remove chemicals and decrease the particle load deposited on the skin surface upon exposure to air pollution, the use of rinse-off products and emollients which reduce transepidermal water loss and increase skin barrier function is recommended [9,51,127]. One group selected E/Z-2-benzylidene-5,6-dimethoxy-3,3-dimethyl-indan-1-one (BDDI), an AhR antagonist, and could show that this component can transiently inhibit AhR activation in human skin in response to air pollution and prevent activation of genes which are relevant for wrinkle formation and skin carcinogenesis such as *CYP1*, *COX-2*, and *MMP-1* [9,130]. This molecule is also hypothesized to be a promising ingredient to prevent senile lentigines formation since it was demonstrated that it can decrease the production of SASP [51]. In general, the lack of testing compounds in advanced models for skin color, photoprotection, and anti-pollution properties is enormous [131]. Especially, the effects of anti-pollution cosmetics on different ethnic skin regarding pigmentation is still missing.

The deposition of environmental stressors on the skin can be prevented by film-forming cosmetics [127]. Lastly, protection against oxidative stress-inducing effects such as ROS generation and downregulation of antioxidant defense mechanisms in the epidermis can be reduced by the application of formulations with antioxidants. The well-known antioxidant Vitamin E has beneficial effects on the skin including decreased expression of Nrf2, restoration of mitochondrial complex I and complex IV, recovery of desmosomal protein integrity, and promotion of cellular migration [57]. Another commonly used antioxidant in cosmetics is l-ascorbic, acid also known as Vitamin C, which can neutralize free radicals, and prevent collagen degradation and hyperpigmentation, which are all signs of skin aging [85,102]. This antioxidant is mostly combined with other antioxidants such as Vitamin E, ferulic acid, l-selenomethionine, and phlorectin and displayed in vivo and in vitro anti-photoaging effects [53,102]. This combination of antioxidants decreases the nuclear translocation of NF-κB and thus promotes the inhibition of inflammation and the synthesis of pro-inflammatory cytokines, respectively [53].

The latest trend in the anti-pollution cosmetic field is the use of ingredients of natural or botanical origin such as algae and green tea leaves. There are several advantages of using naturally occurring ingredients which include increase of cosmetic efficiency and reduction of the risk for allergies and irritation. Furthermore, society demands more ethical, natural, and green formulations in cosmetic research [127]. In conclusion, different components and applications are established which help the skin to prevent or mitigate damages through environmental stress.

## 6. Conclusions

Research on the effects of air pollution on cellular senescence and skin aging has become a popular topic. Therefore, the purpose of this review is to highlight the current understanding of the connections between these topics, and to discuss recent research findings, approaches, and cosmetical interventions used to prevent or mitigate skin damages caused by exposure of the skin to air pollution. Our review demonstrates that cellular senescence and skin aging are closely regulated and connected and that they can, in fact, be induced by air pollution. For this reason, mechanisms of air pollution-induced skin aging have been the focus of many recent publications.

## Figures and Tables

**Figure 1 cells-11-02220-f001:**
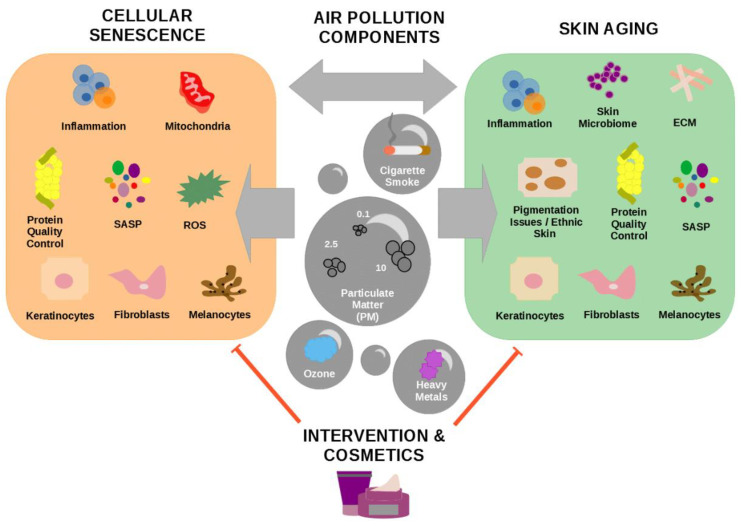
The interplay between air pollution, cellular senescence, and skin aging. The review focuses on the topics inflammation, protein quality control, mitochondrial dysfunction, reactive oxygen species (ROS), and senescence-associated secretory phenotype (SASP) in cutaneous skin cells during senescence induced by air pollution. Additionally, we discuss how air pollution components such as particulate matter (PM), ozone, heavy metals, and cigarette smoke impact skin aging. Finally, we summarize the latest information about interventions and cosmetics which were shown to prevent the impact of air pollution in the processes of cellular senescence and skin aging.

**Figure 2 cells-11-02220-f002:**
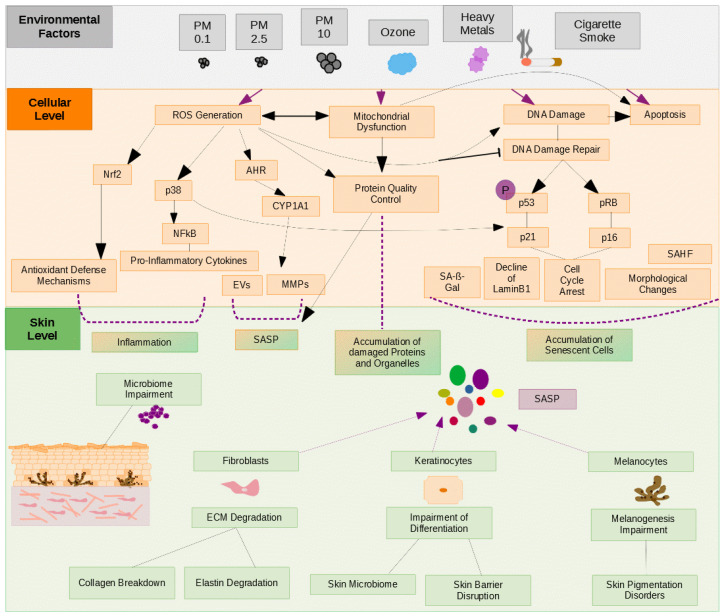
Schematic diagram showing the main signaling pathways involved in the process of cellular senescence which are contributing to skin aging. Pollutants act over many signaling pathways that control cell cycle progression and transcriptional regulation of genes involved in inflammation.

**Table 1 cells-11-02220-t001:** A summary of the current interventions, an example of active ingredients, and which mechanisms are helping against air pollution-induced damage with corresponding references for detailed information.

Intervention	Active Ingredients	Mechanism	References
**Sunscreen**	Oxybenzone Zinc oxide	Prevent synergistic effects of air pollution and UV	[1]
**Washing and air filters**		Prevent deposition and penetration of pollutants on skin	[8,85]
**Dietary habits**	Phenolic compounds in plants	Reduce cellular oxidative stress	[13]
**Rinse-off, film-forming cosmetics, and emollients**	BDDI *Aleurites fordii* oil copolymer Kaolin	Reduced transepidermal water loss Increased skin barrier function	[85]
**Antioxidants**	Vitamin C Vitamin E Ferulic acid	Reduced ROS production and SASP Prevent collagen degradation and hyperpigmentation	[1,85]
**Botanicals**	Algae	Antibacterial and anti-inflammatory activity Skin whitening agent ROS scavenger	[127]

## Data Availability

Not applicable.

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
