# Peer review of "Effects of Air Pollution on Cellular Senescence and Skin Aging"

_cells, 2022, doi:10.3390/cells11142220_

Round 1

Reviewer 1 Report

This is a very comprehensive and interesting review of pollution and its effects on cellular senescence and skin aging. The figures are a very good addition to the text.

I would like to see an additional section on the effects of pollution on ethnic skin types and how this pollution-related skin aging manifests. I feel this would better reflect the move in the dermatology field towards a more inclusive research environment that doesn't only focus on lightly-pigmented skin. 

In the section on cigarette smoke, a recent in vivo study has demonstrated the systemic effect of chronic inhalation of tobacco smoke on the dermis and the elastic fibre components therein and I feel that this paper should be considered for inclusion (Langton et al. J Pathol 2020; 251: 420–428). Similarly, in section 5. green tea catechins have recently been shown in vivo to have a protective effect on the dermal compartment of skin (Charenchon et al. Clin Exp Dermatol. 2022 doi: 10.1111/ced.15179) and I feel this is a study worthy of inclusion. 

These small amendments will enhance this very comprehensive review.

Reviewer 2 Report

The manuscript, a bibliographic review of the effects of pollution on skin senescence and aging, is an extensive description of general cellular processes, some of their particularities in the skin, and, finally, the effects of pollutants.

The manuscript is hard to read; its focus is clear, but it loses during some paragraphs, especially during the first two sections. Some processes deeply described in the first two sections do not contribute to its aim.

I suggest excluding some material to improve the manuscript's readability, maintaining the focus of sections 2 and 3 on the molecular effects that will be discussed in section 4. Figure 1 does not add clarity to the text.

I have some comments to highlight these issues:

  • DNA damage is described on page 3, but just two mentions are present in section 4. Is it necessary to describe the type of damage known if there is no mention of the effects of pollutants? I suggest excluding the types of DNA damage or defining the particular damage induced by pollutants (8-oxo-2'-guanosine, etc.).
  • In a similar way, descriptions of cyclins and CDK are included in section 2, but are absent in section 4 (pollutants' effects). I suggest avoiding information that is not directly involved with the review topic. 
  • On page 3 is described that the lack of proliferation and SA-beta-Gal expression are characteristics of the senescent cells. However, on page 6 is mention: "hallmarks of aging such as cell cycle arrest, decreased autophagy activity, increased SA-β-Gal activity". Are these characteristics hallmarks of aging or senescence? This point is present throughout the manuscript.
  • Page 2: after some general descriptions of senescence authors mention: "This demonstrates how close cellular senescence, skin aging, and skin diseases are connected in terms of molecular processes." No, previous arguments do not demonstrate these facts.
  • Page 6: "In this study the authors have demonstrated that the SASP components of fibroblasts induced to MiDAS promoted keratinocyte differentiation shown by loricrin and transglutaminase 1 expression". How does the keratinocyte differentiation induced by fibroblasts impact skin functions? The only mention of the promotion of KCs differentiation does not represent a clear point in the manuscript. Moreover, if MiDAS is not mentioned in section 4, the entire description of its relationship with senescence extends the manuscript with relatively minor importance for the reader.
  • Page 7: "Activation of the NF-κB signaling pathway is mainly driven by oxidative stress." No, it is not. NF-κB is a master transcription factor activated by a wide variety of immune receptors once they recognize their targets (different TLRs, TNFR, IL-1R, etc.). 
  • Page 7: "The two transcription factors, NF-κB and Nrf2, collab-orate ... to avoid chronic inflammation". How NF-κB, a regulator of inflammation, can avoid chronic inflammation? Which are the mechanisms?
  • The role of Nrf2 is unclear in the manuscript. Page 7 "Pre-treatment ... with EAA prior to UVA irradiation protected cells from ... increased Nrf2 expression." So, Nrf2 increment is an undesired effect. Afterward, the authors mention that "Nrf2 depletion leads to the reduction of cell survival as well as induction of oxidative stress while mutations in the Nrf2 gene, in turn, lead to the development of squamous cell carcinoma suggesting that pathways activated by Nrf2 are important for the maintenance of skin homeostasis." Clearly, Nrf2 expression is a desired effect. How can the authors balance both groups of results?
  • I suggest considering the distant effect produced by PM through immune system activation. It has been described that blocking TNF-alfa after acute exposure to PM reduced cardiac effects (Am J Physiol Heart Circ Physiol. 2015 Nov 15;309(10):H1621-8). TNF-alfa has profound effects on the skin (i.e. psoriasis), so its role cannot be excluded from the manuscript.

Minor comments

There are some errors throughout the manuscript:

  • Page 4: "the nuclear factor kappa-light-chain-enhancer (NF-κB)". The full name of NF-κB is nuclear factor kappa-light-chain-enhancer of activated B cells.
  • Page 5: "In senescent keratinocytes, impairment of autophagy leads to issues in skin differentiation" What do you mean by "issues in skin differentiation"? 
  • Page 5: "During aging, the skin (...) is a useful system for aging studies." Modify the phrase.
  • Page 9: "Skin cells exposed to diesel particulate extract (DPE) ... displayed dysregulation of proteins ... such as NADPH oxidase (NOX), CERAMIDE, plakins, transglutaminases, cystatins, and filaggrin." Ceramide is not a protein.
